# Evolution of Anxiety, Emotional Intelligence, and Effective Personality after COVID-19 among Spanish University Students

**DOI:** 10.3390/bs14030173

**Published:** 2024-02-23

**Authors:** Andrea González Rico, Cristina Di Giusto Valle, María-Camino Escolar-Llamazares, Tamara de la Torre Cruz, Isabel Luis Rico, María Eugenia Martín Palacio

**Affiliations:** 1Facultad de Educación, Universidad de Burgos, 09001 Burgos, Spain; andrea.gonric@educa.jcyl.es (A.G.R.); tdtorre@ubu.es (T.d.l.T.C.); miluis@ubu.es (I.L.R.); 2Facultad de Ciencias de la Salud, Universidad de Burgos, 09001 Burgos, Spain; 3Facultad de Educación, Universidad Complutense de Madrid, 28040 Madrid, Spain; mariaeugeniamartin@edu.ucm.es

**Keywords:** anxiety, COVID-19, emotional intelligence, effective personality, university students

## Abstract

The repercussions of the pandemic caused by the SARS-CoV-2 coronavirus over recent years have posed an unprecedented challenge for the whole of society, affecting the well-being of everyone. Among all the variables affected in relation to well-being, Anxiety, Emotional Intelligence, and Effective Personality (Self-Esteem, Academic Self-Realisation, Resolute Self-Efficacy, Social Self-Realisation) have been highlighted. The aim of this study is to assess the evolution of those variables across three temporal phases: pre-pandemic, during the pandemic, and up until the end of the study in April 2022. A study was conducted during these temporal phases with three cohorts from Spanish Universities. The cohorts were formed of people assessed for Anxiety (660 pre-pandemic, 460 during the pandemic, and 311 at the end of the study), Emotional Intelligence (355 pre-pandemic, 91 during the pandemic, 311 at the end of the study), and Effective Personality (708 pre-pandemic, 174 in 2018, 311 at the end of the study). Anxiety was assessed with the State–Trait Anxiety Inventory, Emotional Intelligence with the Trait Meta-Mood Scale and TMMS-24, and Effective Personality with the Cuestionario Personalidad Eficaz-Universidad (the Effective Personality Questionnaire-University). The results showed a rise in the state of anxiety during COVID-19, with a subsequent reduction two years into the pandemic; however, anxiety rates remained higher than before the pandemic. Emotional intelligence increased in the emotional attention factor, but diminished as regards both clarity and regulation. Effective Personality was at lower levels for all factors (Self-Esteem, Academic Self-Realisation, Resolute Self-Efficacy, Social Self-Realisation). The main conclusion was the need for assistance with the dimensions under study, in order to improve the well-being of university students after the serious effects caused by COVID-19.

## 1. Introduction

The coronavirus SARS-CoV-2 pandemic has over the past three years become one of the worst global health threats in the history of humanity. Its impacts across the entire planet have set an unprecedented challenge for society as a whole [1]. The WHO [2] recommended, among other matters, that countries should order the confinement of the public to mitigate the effects of the pandemic. Among the negative impacts of those measures, the adverse consequences for the general well-being of individuals have been foremost [3].

With the above in mind, Soto & Zuñiga [4] and Velastegui et al. [5] found that, during the pandemic, all of society expressed problems related to anxiety, fear, stress, loneliness, and sadness. At the international level, they showed a significant increase in the prevalence of depressive and anxiety-related disorders attributable to COVID-19.

Likewise, Salari et al. [6] in a recent study reported that, due to COVID-19, the population has presented depression, changes of mood, emotional distress, irritability, post-traumatic stress, insomnia, and rage. They offered evidence of individuals experiencing panic attacks, psychosis, anxiety, and even suicidal thoughts, as well as emotions such as sadness, fear, and rage.

More specifically, university students are seen as an especially vulnerable group that suffered during the pandemic. They are not only in the process of transition from adolescence to adulthood, but they also have to struggle with the academic stress generated by their studies, both processes being seen as stress factors. In fact, on comparing these students with other age groups, it was found that their levels of stress and depression were so much higher that they even presented clinical and subclinical psychological symptomatology [5]. As regards differential behaviour by sex, Molero-Jurado et al. [7] pointed to higher levels of anxiety among women.

Because of all the difficulties experienced after the pandemic, the aim of the present research is centred on discovering the evolution of the variables of Anxiety, Emotional Intelligence, and Effective Personality over three temporal phases (pre-pandemic, during the pandemic, and up until the end of the study in April 2022). The specific objectives were as follows:

O1. To discover the evolution of the variables of Anxiety throughout the three temporal phases (pre-pandemic, confinement or lockdown, and the end of the study) and to check whether there are differences between the genders of the participants.

O2. To trace the evolution of the variables of Emotional Intelligence throughout the three temporal phases (pre-pandemic, confinement or lockdown, and the end of the study) and to check whether there are differences between the genders of the participants.

O3. To trace the evolution of the variables of Effective Personality throughout the three temporal phases (pre-pandemic, confinement or lockdown, and the end of the study) and to check whether there are differences between the genders of the participants.

O4. To discover the relations between Anxiety, Emotional Intelligence, and Effective Personality.

From the above objectives, the following hypotheses were proposed:

**H1.** 
*Levels of Anxiety have diminished following the pandemic, but are still higher than before it began, both among men and among women.*


**H2.** 
*Levels of Emotional Intelligence have increased after confinement, but are lower than before the pandemic, both among men and among women.*


**H3.** 
*Levels of Effective Personality have diminished as against the pre-pandemic levels, both among men and among women.*


**H4.** 
*Anxiety will be negatively correlated with the major variables of Emotional Intelligence and Effective Personality, while Emotional Intelligence will be positively correlated with Effective Personality.*


### Literature Review

In terms of quality of life, it has been found that Anxiety-related disorders influence the normal development of adolescents, presenting adverse repercussions for both psychosocial performance and academic output [8]. We shall consider anxiety as defined by Hernández et al. [9], who described it as the response of the organism to a concrete stimulus, which triggers a reaction that activates an individual’s central nervous system. Within the university educational environment, it has been noted that an ever-greater number of students present emotional problems related to anxiety, which affects between 13% and 17% of students. As regards the state factor, for Spielberger [10], state anxiety is an immediate “emotional state” modifiable through time. As regards anxiety, it is seen as a psychological disorder that affects the population in general, but especially in the university context. Students at that stage of education, throughout their life as students, face varying levels of anxiety, which can be considered a normal part of student life [11].

In relation to gender and different levels of anxiety in the university population, several authors have indicated a different rate of maladaptive anxiety between 23–25% in men and between 36–42% in women [11,12,13], a difference that is two to three times higher than the general population (8% and 15%, respectively). However, others found no such differences [14].

Emotional Intelligence is defined by Salovey & Mayer [15] as “the ability to monitor one’s own and others’ feelings and emotions, to discriminate among them, and to use this information to guide one’s thinking and actions” (p. 187). This study is based on the construct of Emotional Trait Intelligence. Emotional Trait Intelligence encompasses behavioural tendencies and self-perceptions related to the ability to identify, process, and use emotionally charged information within various contexts, remaining aware of individual differences [16,17]. Authors such as Joseph & Newman [18], Filippello, et al. [19] supported the idea that Emotional Trait Intelligence is conceptually different from Emotional Intelligence. The latter is considered as a series of cognitive skills related to emotions that imply capacity, while a trait refers to the self-perceptions of person’s own emotional abilities, which is why they involve different processes [20,21,22].

Emotional Intelligence has been extensively studied in university populations. In the study of Fragoso [23], almost half of the students needed to improve their emotional attention skills (45%), 42% emotional clarity, and 31% emotional repair. The Emotional Intelligence of university students is also related to different variables such as the studies that relate it to coping strategies for stress [24], negative affect related to suicide among young people, and performance [25].

Following Barrera & Flores [26], the construct of emotional intelligence between the sexes and its differences could be explained through gender roles, as regards the expectations that may be assumed or attributed to women. Those expectations could include a natural ability in caring for others, amiability, accessibility, and, in general, characteristics that favour social exchange and interaction.

In relation to the Effective Personality, Martín del Buey et al. [27] (p. 34) indicated that an effective person is a living being with knowledge of, and esteem for, the self (self-concept and self-esteem); in a process of ongoing maturation (in any state of its evolution); with capacity (intelligence) for achieving (effectiveness) what the person desires (motivation) and hopes to achieve (expectation); using for that purpose the best means (training) possible (efficiency); controlling the causes (attribution of causality) of any achievement (success or failure); confronting personal, circumstantial, and social difficulties (confronting problems) as they may arise; taking balanced decisions with neither detriment to good relations with others (empathy and communication); nor renouncing fair personal aspirations (assertiveness) in their decisions.

Padilla [28] emphasised that over a third of the 68 university students who formed his sample showed low levels of Effective Personality, and thus presented a counterproductive prognosis; a result that was in agreement with Roldán [29], who indicated that appropriate self-esteem and self-concept are beneficial for individual development, as they provide tools to deal with the stressful elements of the environment and thus generate a feeling of well-being. It was observed that COVID-19, as a stressor, affected the selected variables, which are defined below in what follows.

In relation to gender-based differences and Effective Personality, authors such as Castellanos et al. [30] indicated that women had higher scores for Academic Self-Actualisation; while men showed better Self-Esteem and Resolutional Self-Efficacy. While other studies such as that of Vargas [31] found no such differences among the university population.

Different studies have shown the existence of a direct and significant relationship between the variables of those constructs. For example, close relations have also been found mainly between Anxiety and Emotional Intelligence, Anxiety and Effective Personality, and Emotional Intelligence and Effective Personality, although there is no evidence linking all three together.

Specifically, a negative and statistically significant relation was found between the variables of Emotional Intelligence: emotional understanding and regulation [32]. According to Bojórquez & Moroyoqui [33], most students presented levels of emotional intelligence varying from adequate to excellent, while levels of both state and trait anxiety were between medium and high.

Positive and statistically significant relations have been found between emotional attention and both state anxiety [33,34] and trait anxiety [34]. Statistically significant relations, but in the negative sense, have also been found between emotional clarity and state anxiety [33,35], and likewise with trait anxiety [35]. In the same sense, a negative relation has been noted between emotional regulation and both state and trait anxiety [33]. It all concurs with the studies of Thomas et al. [36], who also observed a statistically significant negative relation between Anxiety-related and Emotional Intelligence-related factors, attributing the results to the fact that one or more of the emotional-processing variables were defective.

Okwuduba et al. [37] found a significant correlation between interpersonal and intrapersonal intelligence, self-directed learning, and the academic output of university students. The results of the study carried out by Ramos et al. [38] showed a statistically significant and positive relation between the emotional skills of individuals and their levels of empathy. The relation of Effective Personality with stress has been indicated by Tirado [39], who pointed out that people with low self-esteem rarely possess the necessary capacities for decision-making, conflict resolution, and social skills. The likelihood that they will think of themselves as incapable will therefore increase when facing academic challenges, which will in turn generate high levels of academic stress.

Relating the constructs of Effective Personality and Emotional Intelligence, Tapia [40] discovered the existence of statistically significant and positive correlations between the dimensions of both constructs, except for the relations factor, which was not found to correlate with the dimensions of Emotional Intelligence.

Once the variables had been defined, and having reviewed the literature in which relations were established between each one, we proceeded to describe the impact of COVID-19 upon them. In a study carried out in the context of the pandemic, regarding the relation between Emotional Intelligence and Anxiety, a greater level of correlation was found among women than among men [41]. Other studies, such as those of Azañedo [42] and Crisostomo [43], regarding Emotional Intelligence and self-esteem after the pandemic, indicated that 80.0% of the participants presented medium levels of emotional intelligence, and 50.8% had medium levels of self-esteem.

Maladjustment to physical self-concept have also changed, due to governmental decrees, which have urged both social isolation and the quarantines imposed at an international level, with the aim of reducing levels of coronavirus infection [44]. Along those lines, Wang et al. [45] pointed out that the forced isolation that people have lived through has had a negative impact on self-esteem, which is related to aspects such as anxiety, depression, and social skills [46].

## 2. Methods and Materials

In this study, a quantitative methodology was employed, by means of a temporal-interval design throughout three temporal phases (pre-pandemic, during the pandemic, and up until the end of the study).

### 2.1. Participants

Different temporal phases were used for this research: pre-pandemic, pandemic, and present-day samples (with the exception of Effective Personality, for which no data were found for the pandemic period; a sample from 2018 was used, in order to have knowledge of its general tendency over recent years).

#### 2.1.1. Pre-Pandemic Phase

For the pre-pandemic samples, the rating scales for State Anxiety and Effective Personality as validated instruments were used. Regarding Emotional Intelligence, data were taken from a scientific article with a Spanish university sample, as it is a widely used instrument, but it has no official validated rating scale.

-Anxiety

For this sample, we used the participants who made up the sets of tables for state anxiety of the State–Trait Anxiety Inventory (STAI) of Spielberger et al. [47] in Spanish adults. The samples comprised 660 adults, of whom 295 (44.70%) were men and 365 (55.30%) were women.

-Emotional Intelligence

The study of Cazalla-Luna and Molero [48] was used, as it is the most relevant article in the Web of Science (with the keyword search terms TMMS and University and Spain). The article is the one with most references and has the largest sample for Spanish universities. The participants were 355 university students on education-related courses at the University of Jaén. Sample distribution by gender was 75 (21.13%) men and 280 (78.87%) women, with an average age of 23.78 years (SD = 5.55).

-Effective Personality

For this sample, the participants who formed the rating scale of the Effective Personality questionnaire [49] in Spanish universities were used. The sample was made up of 708 adults, of whom 210 (29.66%) were men and 498 (70.34%) women, with an average age of 21.15 years (SD = 3.64).

#### 2.1.2. Pandemic Phase

The data available in the articles published in the Web of Science were selected for each temporal phase under study; these variables had been studied during confinement in the university population, using the same instruments, with the exception of Effective Personality, as previously mentioned.

-Anxiety

The sample in the study of Garcia-Gonzalez [50] was used, formed of 460 university degree nursing students living in the autonomous communities of Murcia and Almería. The gender distribution of the sample was 101 (21.96%) men and 359 (78.04%) women, with a mean age of 20.58 (SD = 1.54). Anxiety was evaluated in the first and fourth week of confinement.

-Emotional Intelligence

The largest sample of Spanish university students was found in the study of Sánchez-Cabrero et al. [51], the most relevant article in the Web of Science (with the search terms TMMS and University and COVID-19) and with the most references. The participants comprised 91 university students. The gender distribution was 32 (35.16%) men and 59 (64.84%) women, with an average age of 35.9 years (SD = 7.88). Emotional Intelligence was assessed from September 2020 to July 2021.

-Effective Personality

No data were found during the pandemic in Spanish universities, and so it was decided to use a sample close to the pandemic, as an indication of its evolution during the previous period. The sample in the study of Serrano [52] was used, comprising 174 university students on Health Science courses at the Complutense University of Madrid. The distribution by gender was 25 (14.4%) men and 149 (85.6%) women, with a mean age of 22.44 years (SD = 2.60).

#### 2.1.3. Post-Pandemic Phases

The three questionnaires were applied to the most recent sample compiled in March and April 2022. Convenience sampling and one case of snowball sampling were used to select the participants. The total sample was formed of 311 Spanish university students. The gender distribution of the sample was 49 (15.76%) men and 262 (84.24%) women, and their average age was 22.02 years old (SD = 3.19).

In summary, Table 1 presents the data on the participants for both men and women in each phase.

### 2.2. Instruments

Three questionnaires for exploring Anxiety, Emotional Intelligence, and Effective Personality were used in the study:

State–Trait Anxiety Inventory (STAI E/R) [10,46]. The STAI consists of a total of 40 items: the first twenty measure State Anxiety and the last twenty measure Trait Anxiety. It presents an internal consistency reliability of 0.90–0.93 and features a 3-point Likert-type scale of 0 to 3 where 0 indicates none and 3 a lot.

Trait Meta-Mood Scale (TMMS-24) for measuring Emotional Intelligence [53]. An adaptation to the Spanish population of the Trait Meta-Mood Scale [15] it consists of 24 items, divided into three dimensions, each of eight items, which are: Emotional Attention, Emotional Clarity/Comprehension, Emotional Regulation/Repair. This instrument presents a Cronbach’s alpha reliability of 0.90. The internal consistency of each subscale was over 0.85 in all cases, and the test–retest correlations after 4 weeks ranged between 0.60 to 0.83. The items were scored on a five-point Likert-type scale, ranging from 1 = zero agreement to 5 = complete agreement, so the score ranges of each dimension were between one and forty.

University Effective Personality Questionnaire (CPE-U) [49]. This instrument consists of 30 items and presents a Cronbach’s alpha reliability of 0.877 in the Spanish sample. The questionnaire is answered on a five-point Likert-type scale with scores from 1 = never to 5 = always. It evaluates four dimensions of Effective Personality: Self-Esteem (ego-Strengths); Academic Self-Realisation (dealing with Demands); Resolute Self-Efficacy (rising to Challenges); Social Self-Realisation (Personal relations).

### 2.3. Procedure

This study was developed during the second semester of the 2021–2022 academic course. After reviewing the literature, the aims, the hypotheses, and instruments of the study were defined. Subsequently, the articles describing pre-pandemic and pandemic phases were selected, based on the previously mentioned participant selection criteria. Finally, the questionnaire was administered to the set of Spanish universities to obtain the post-pandemic phase participants. The questionnaires were applied on a voluntary basis in the Microsoft Forms format, thereby ensuring confidentiality and anonymity of the information that was gathered, and expressing the statistical aim of the results that were obtained. The need for sincerity on the part of the participants was pointed out, as was the inapplicability of any notions of whether the replies could ever be either correct or incorrect. In case of any doubts, the students were individually contacted through email.

The study was performed in accordance with the ethical standards as laid down in the 1964 Declaration of Helsinki and its later amendments or comparable ethical standards. Furthermore, the students were informed of the objective of the investigation, procedures and the anonymised treatment of all data. They were also told that their participation was entirely voluntary, and they could abandon the research project any time without any prejudicial consequences. All participating students signed an informed consent form. All data collected for this study have undergone systematic anonymisation to protect any personal identification. The anonymisation protocol was implemented following the instructions of the Ethics Committee of the University of Burgos.

### 2.4. Data Analysis

The summary-data T-test for independent samples with Bonferroni correction (0.05/n tests) was used in the three temporal phases, with the aim of meeting the three first objectives of knowing the evolution of Anxiety, Emotional Intelligence, and Effective Personality. This test was used, due to the robustness of the Student *t*-Test against any violation of the assumptions of independence, normality, homogeneity of variances, and sampling randomness [54]. As regards the fourth objective of exploring the relation between variables, the Spearman Rho correction test was used, as one or another of the assumptions of normality, linearity, or heteroscedasticity, required for performing Pearson’s parametric correlation test, had not been met. All analyses were carried out with the SPSS 25.0 program.

## 3. Results

The first objective was to discover the evolution of State Anxiety in the three phases (pre-pandemic, confinement, end of study). Figure 1 shows that the levels of anxiety increased drastically in the first week of confinement and even more in the fourth week. After two years of the pandemic, the values had returned to similar, although slightly higher pre-pandemic levels.

On analysing whether there were statistically significant differences between the separate phases, it was found that both men (M = 23.45; SD = 12.50) and women (M = 26.69; SD = 11.70) showed significantly lower levels of state anxiety at the end of the study than in the fourth week of confinement (men with M = 58.50 and SD = 10.60; women with M = 60.30 and SD = 10.30).

On comparing the results with the first week of quarantine, it was also found that both men (M = 23.45; SD = 12.50) and women (M = 26.69; SD = 11.70) had reached significantly lower levels (*p* < 0.017) of state anxiety at the end of the study than in the first week of confinement (men with M = 44.70 and SD = 19.40; women with M = 52.00 and SD = 20.90).

Finally, on comparing the results with the rating scale (for the pre-COVID-19 test) it was found that, in the present period, men (M = 23.45; SD = 12.50) had indeed recovered pre-COVID-19 levels (M = 20.54 and SD = 10.56), there being no significant difference between their levels (*p* > 0.017). Women, on the other hand, continued to show significantly higher scores (*p* < 0.017) in their levels of state anxiety with respect to the pre-COVID-19 values (M = 23.30 and SD = 11.93).

The second objective was to discover the evolution of Emotional Intelligence (Attention, Clarity, and Regulation) in the three phases (pre-pandemic, during the pandemic, and end of the study).

In Figure 2, it can be seen that the Attention levels of Emotional Intelligence increased among women during the pandemic and continued to increase slightly more until the end of the study, whereas they were maintained among men during the pandemic and even continued to increase slightly.

On analysing whether statistically significant differences existed between the different moments, it was found that both men (M = 26.80; SD = 7.05) and women (M = 29.04; SD = 6.11) showed only a slight increase, without presenting statistically significant differences (*p* > 0.025), at the end of the study, over their levels of attention during the pandemic (men with M = 26.19 and SD = 6.89; women with M = 28.63 and SD = 5.93).

On comparing the results with the pre-COVID-19 scores, it was found that men (M = 26.80; SD = 7.05) showed an insignificant improvement (*p* > 0.025) at the end of the study over their pre-pandemic levels (M = 26.28 and SD = 5.58), whereas women (M = 29.04; SD = 6.11) presented significantly higher scores (*p* < 0.025) in their attention levels with respect to their pre-pandemic values (M = 27.33 and SD = 5.17).

In Figure 3, we can see that levels of the clarity factor of Emotional Intelligence increased in women during the pandemic, but descended afterwards, remaining at a lower level than at the pre-pandemic stage. On the other hand, men’s scores diminished during the pandemic and were at a slightly lower level at the end of the study.

On analysing whether there were statistically significant differences between the different temporal phases, it was found that men (M = 24.82; SD = 6.52) had not worsened significantly (*p* > 0.025) at the end of the study, as compared with their scores during the pandemic (M = 25.78 and SD = 6.68), while women (M = 24.51; SD = 6.67) presented lower scores at the end of the study than during the pandemic period (M = 29.58 and SD = 5.68).

Comparing the results with the pre-COVID-19 scores, it was found that both men (M = 24.82; SD = 6.52) and women (M = 24.51; SD = 6.67) had significantly lower scores (*p* < 0.025) at the end of the study with respect to their pre-COVID-19 levels (men with M = 27.56 and SD = 4.97; women with M = 27.64 and SD = 5.14).

In Figure 4, we see that the levels of regulation of Emotional Intelligence among women increased during the pandemic but fell drastically at the end of the study. Among men, on the other hand, they fell drastically during the pandemic, since when they have slightly risen.

Upon analysing whether there were statistically significant differences between the different temporal phases, it was found that men (M = 25.90; SD = 6.67) had improved slightly but not significantly (*p* > 0.025) at the end of the study in comparison with the pandemic (M = 25.06 and SD = 6.70). Nonetheless, the levels of regulation of women had significantly worsened (*p* < 0.025) at the end of the study (M = 23.63; SD = 6.08) in comparison with during the pandemic (M = 27.42 and SD = 6.23).

Finally, on comparing the results with the pre-pandemic scores, it was found that men (M = 25.90; SD = 6.67) had indeed recovered pre-pandemic levels (M = 28.13 and SD = 5.45) at the end of the study, there being no significant differences between the scores (*p* > 0.025). Women (M = 23.63; SD = 6.08) continued to show significantly lower scores (*p* < 0.025) in their levels of regulation at the end of the study compared to their pre-COVID-19 values (M = 27.13 and SD = 5.38).

The third objective was to discover the evolution of Effective Personality (Self-Esteem, Academic Self-Realisation, Resolute Self-Efficacy, and Social Self-Realisation) over the three phases (pre-pandemic, during the pandemic, and the end of the study).

Figure 5 shows us that the levels for Self-Esteem of Effective Personality had increased slightly among women in 2018 and fell towards the end of the study, while for men they continued to fall slightly during both phases (pandemic, the end of the study).

On analysing whether there were statistically significant differences between the separate phases, it was found that men (M = 27.88; SD = 5.21) maintained similar levels to those of 2018 at the end of the study (M = 28.92 and SD = 4.46), and no significant differences were found between them (*p* > 0.025), whereas women (M = 24.54; SD = 5.90) showed significantly lower scores (*p* < 0.025) in their levels of Self-Esteem at the end of the study in comparison with the 2018 values (M = 29.48 and SD = 4.38).

On comparing the results with the rating scales (pre-pandemic), it was found that men (M = 27.88; SD = 5.21) maintained the pre-pandemic level (M = 29.06 and SD = 4.70), no significant differences being found between them (*p* > 0.025). On the other hand, women (M = 24.54; SD = 5.90) showed significantly lower scores (*p* < 0.025) in their levels of Self-Esteem at the end of the study in comparison with the pre-pandemic values (M = 27.19 and SD = 4.92).

In Figure 6, we can see that the levels for Academic Self-Realisation of Effective Personality increased both among men and among women in 2018 but fell drastically towards the end of the study.

On analysing whether statistically significant differences existed between the different temporal phases, it was found that both men (M = 26.69; SD = 4.69) and women (M = 27.43; SD = 5.42) showed significant decreases (*p* < 0.025) in their levels of Academic Self-Realisation at the end of the study as against 2018 (men with M = 31.36 and SD = 4.20; women with M = 31.48 and SD = 4.19).

Finally, on comparing the results with the test rating scales (pre-pandemic), it was found that men’s scores (M = 26.69; SD = 4.69) had, at the end of the study, fallen to the pre-pandemic level (M = 28.30 and SD = 4.53), with no significant differences between them (*p* > 0.025). Women, on the other hand (M = 27.43; SD = 5.42), showed significantly lower scores (*p* < 0.025) in their levels of Academic Self-Realisation at the end of the study than their pre-pandemic values (M = 29.20 and SD = 3.93).

In Figure 7, we can see that the levels for Resolute Self-Efficacy of Effective Personality increased both among men and among women during 2018, but had fallen at the end of the study, with the changes being most notable among women.

On analysing whether statistically significant differences existed between the different temporal phases, it was found that the men’s scores (M = 17.02; SD = 3.36) were slightly lower at the end of the study in comparison with their scores for 2018 (M = 18.48 and SD = 2.60), without the differences being significant (*p* > 0.025). Women, on the other hand (M = 15.97; SD = 3.34), presented significantly lower scores (*p.*< 0.025) for their levels of Resolute Self-Efficacy at the end of the study than their values for 2018 (M = 18.75; SD = 2.57).

When comparing the results with the test rating scales (pre-pandemic), it was found that both men (M = 17.02; SD = 3.36) and women (M = 15.97; SD = 3.34) showed significantly lower levels for Resolute Self-Efficacy at the end of the study with regard to the values on the rating scales (men with M = 18.19 and SD = 3.23; women with M = 17.27 and SD = 3.07).

In Figure 8, can see that the levels for Social Self-Realisation of Effective Personality have been gradually falling for both men and women, but less so among women than among men.

On analysing whether statistically significant differences existed between the different temporal phases, it was found that men’s scores for Social Self-Realisation (M = 33.96; SD = 6.85) had fallen slightly since 2018 (M = 34.80 and SD = 5.28), but that those differences were not statistically significant (*p* > 0.025). On the other hand, women (M = 33.19; SD = 6.37) showed significantly lower scores (*p* < 0.025) for Social Self-Realisation at the end of the study than in 2018 (M = 35.20 and SD = 5.72).

Finally, when we compared the results with the test rating scales (pre-pandemic), we found that men’s scores (M = 33.96; SD = 6.85) were at a lower level at the end of the study than before the pandemic (M = 36.01 y SD = 5.74), without observing statistically significant differences between them (*p* > 0.025). However, women (M = 33.19; SD = 6.37) showed significantly lower scores (*p* < 0.025) for Social Self-Realisation at the end of the study compared with the pre-pandemic values (M = 36.34 and SD = 5.25).

Finally, as regards the fourth objective of discovering the relations between Anxiety, Emotional Intelligence, and Effective personality, the Spearman’s Rho test was performed (Table 2). It was observed that anxiety was negatively correlated with Emotional Intelligence and Effective Personality among both men and women, with the exception of the Attention factor of Emotional Intelligence. The latter was also positively correlated with Effective Personality, with the exception of the Attention factor, which was not correlated with any other factor among men and was only correlated with Academic Self-Realisation and Resolute Self-Efficacy among women; neither was the clarity factor, nor the regulation factor, correlated with Academic Self-Realisation among the group of men.

## 4. Discussion

In the following discussion, the first hypothesis, “Levels of Anxiety have diminished following the pandemic, but are higher than before it, both among men and among women”, was only partially supported, since it proved true for women, but not for men, whose anxiety levels had already reverted to normal levels.

These results led to conclusions that were supported in studies such as those by Soto & Zuñiga [4] and Velastegui et al. [5], who pointed out that the university population presented problems of anxiety, evidencing a statistically significant increase in the presence of anxiety disorders during COVID-19, since, compared to other social groups, university students were those that suffered the highest levels of anxiety. On similar lines, Cullen et al. [55], Liyanage et al. [56], and González et al. [57] stated that high levels of indicators of anxiety and stress were observed during the pandemic. Likewise, Tijerina et al. [8] added that anxiety disorders affected normal development, causing a negative impact on the psychosocial performance of students.

With regard to gender differences, Molero-Jurado et al. [7] pointed to higher levels of anxiety among women than among men. Other recent studies have made it clear that the population has presented depression, irritability, changes of humour, emotional anguish, insomnia, post-traumatic stress, and anger [6] due to COVID-19. Evidence has also been shown that individuals have experienced psychosis, panic attacks, anxiety, and even suicidal thoughts, besides emotions such as fear, sadness, and rage. It could therefore be said that anxiety levels were higher, because the demands and self-demands of present-day society are much greater for women than they are for men, above all in the area of universities.

As regards the second hypothesis, “Levels of Emotional Intelligence have increased after confinement, but are lower than before the pandemic, both among men and among women”, we can see that it is partially fulfilled, because, taking the different factors into account, attention levels increased during the pandemic and have continued to increase up until the end of the study, while among men they increased during confinement and have continued with slight increases.

As for clarity, levels increased among women during the pandemic, but fell afterwards, being at present below pre-COVID-19 levels, while men’s scores fell progressively up until the end of the study. As for the levels for Regulation, women showed an increase during the pandemic, while men’s scores fell significantly during the pandemic and have since risen. These results are supported by various studies, among which the study of Bojorquez and Moroyoqui [33], in which they found that the majority of university students had levels of emotional intelligence ranging from adequate to excellent. Therefore, emotionally intelligent individuals ought to be able to field more effective strategies for dealing with problems and express reactions and responses to stress levels of lesser intensity [58]. Aguirre [59] emphasised emotional training as a tool that facilitates the resolution of specific problems through knowledge, skills, forms of behaviour, and values.

As regards the results, the differences between the sexes could be explained, along the lines of Barrera and Flores [26], through gender roles, such as the expectations assumed by or attributed to women, i.e., caring for others, amiability, accessibility, and, in general, characteristics that favour exchange and social interaction. Likewise, during the pandemic, several types of behaviour appeared that were situated outside the traditional stereotypes.

The results confirmed the third hypothesis, “Levels of Emotional Intelligence have diminished as against pre-pandemic levels, both among men and among women”, since levels of Emotional Intelligence fell progressively as against the pre-pandemic levels for all factors. The results can be contrasted with the work of Padilla [28] who found during his research that 35% of the students presented a low level of Effective Personality in his samples. These results are in agreement with those of Huang et al. [60], who found that university students showed little motivation in their daily routines during the quarantine phase, with headaches and lack of control, which had repercussions on their levels of both anxiety and depression. Results which were corroborated in the study of Vásquez et al. [61], which evidenced high levels of anxiety and depression.

With regard to the contributions of Roldán [29], the fact that students had adequate levels of self-concept and self-esteem meant that they had the necessary tools to confront the stresses of surrounding pressures, and in turn to give them feelings of well-being. Thus, it may be that the scores rose during the pandemic period because, despite the anxiety that was generated, people were able to dedicate more time to themselves and, especially among women, the tendencies for introspection and personal care were stimulated, in both personal and interpersonal competences.

The fourth hypothesis, “Anxiety will be negatively correlated with the factors of Emotional Intelligence and Effective personality; while Emotional Intelligence will be positively correlated with Effective personality” was partially confirmed, since Anxiety was negatively correlated with the major variables of Emotional Intelligence and Effective personality, although the same correlation was missing between Anxiety and the attention factor of Emotional Intelligence, and between Emotional Attention and Effective personality in men. These results contradicted the findings of Bojórquez and Moroyoqui [33] and Botero and Delfino [34], in which a positive relation was noted between state anxiety and emotional attention, since in the latter study no correlation was found either among men or among women. Nevertheless, the results of our study were in agreement with them and with De Ávila et al. [32] and Thomas et al. [36], in so far as a negative relation was noted between Anxiety and emotional clarity and regulation. On analysing the relation between Anxiety and Effective Personality, it was observed that the results were in partial agreement with those of Baeza [62], who also found a negative relation between anxiety and Self-Esteem, Academic Self-Realisation and Resolute Self-Efficacy, but not with Social Self-Realisation; a relation that was found in this study.

Finally, Emotional Intelligence was positively correlated with Effective Personality, except for the attention factor, which did not correspond with any other factor in men, and only with the Academic Self-Realisation and Resolute Self-Efficacy among women; neither the clarity factor nor the regulation factor was correlated with Academic Self-Realisation in the male group. A result that was in partial agreement with the results of the study by Tapia [40], who found that Effective Personality correlated positively with all the dimensions of Emotional Intelligence. It was also partly supported by Crisostomo [43], who noted a positive correlation between Emotional Intelligence and self-esteem, and Ramos et al. [38] when indicating the existence of positive correlations between emotional competences and levels of empathy.

## 5. Conclusions

As regards the objectives set out in this study, we can firstly conclude that although anxiety values have increased significantly during COVID-19, they have returned to pre-pandemic values among men but have remained higher among women.

In second place, an increase in women’s scores in all dimensions of Emotional Intelligence was observed during the pandemic. But this increase has reverted to values even lower than pre-pandemic at the time of this study, except for emotional attention that has continued to increase. Regarding men, a slight, but not significant, increase in emotional attention scores has been seen over time. Despite this, they have experienced a decrease during the pandemic in the other two dimensions. This decrease has been maintained in emotional clarity but has been reversed, reaching values similar to pre-pandemic levels for emotional regulation.

Thirdly, the levels of Effective Personality had a disparate evolution. Regarding Self-Esteem, it was observed that men have maintained it in a similar way during all periods, while women, after the pandemic, have significantly decreased their scores. In Academic Self-Realisation and Resolute Self-Efficacy, an improvement in scores was observed in 2018 and a considerable decrease after the pandemic, among both men and women. In relation to the Social Self-Realisation, a progressive decrease occurred in both sexes.

And finally, it was found that anxiety was negatively correlated with the factors of Emotional Intelligence and Effective Personality, while Emotional Intelligence was positively correlated with Effective Personality. The dimension of emotional attention was an exception, which was neither correlated with anxiety nor, among the group of men, with Effective Personality.

Regarding the limitations found in the present study, reference can be made to the small and unequal numbers of participants in some of the temporal phases. Another limitation was not to have found other studies on Effective Personality carried out during the pandemic period, so that the evolution of those variables could not be contrasted for pandemic conditions. Likewise, the fact of having used studies found in the Web of Science (on account of these papers being regarded as of maximum scientific seriousness), leaving aside other studies where the samples had more affinity with those of our research, could affect the comparison that was drawn. Hence, as a future line of research, other databases could be taken into account for the above analysis. Nor have earlier temporal phases been found in which all the variables studied here were analysed as a whole.

Due to the generalised deterioration in all the variables under analysis and the relations found among them, the need arises, as a future line of action, to implement intervention programs focused on developing some social or emotional capacities within the students. These strategies could be useful for dealing with Anxiety, and improving Emotional Intelligence and Effective Personality, since these have an impact on personal well-being. Another possible future line of inquiry is the replication of the study with other populations and/or other countries.

## Figures and Tables

**Figure 1 behavsci-14-00173-f001:**
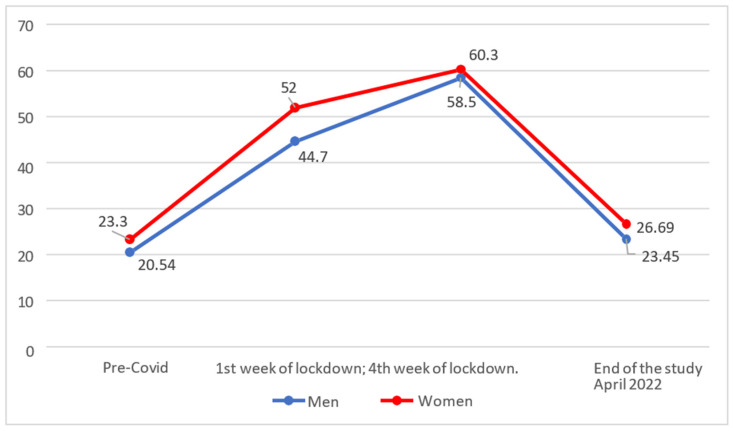
Evolution of State Anxiety.

**Figure 2 behavsci-14-00173-f002:**
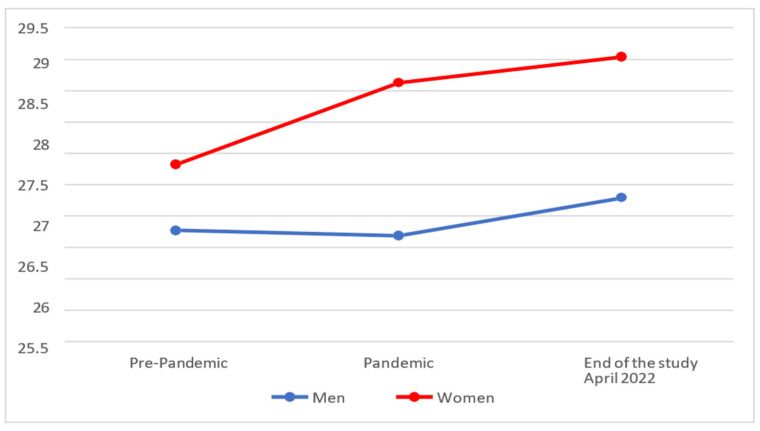
Evolution of the Attention factor of Emotional Intelligence.

**Figure 3 behavsci-14-00173-f003:**
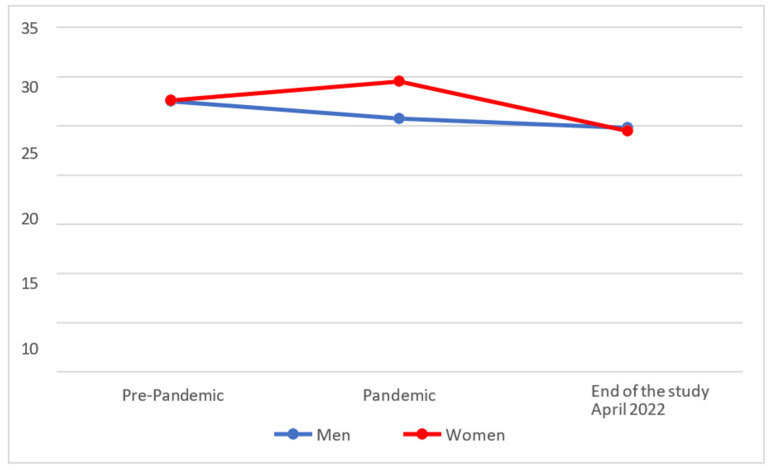
Evolution of the clarity factor of Emotional Intelligence.

**Figure 4 behavsci-14-00173-f004:**
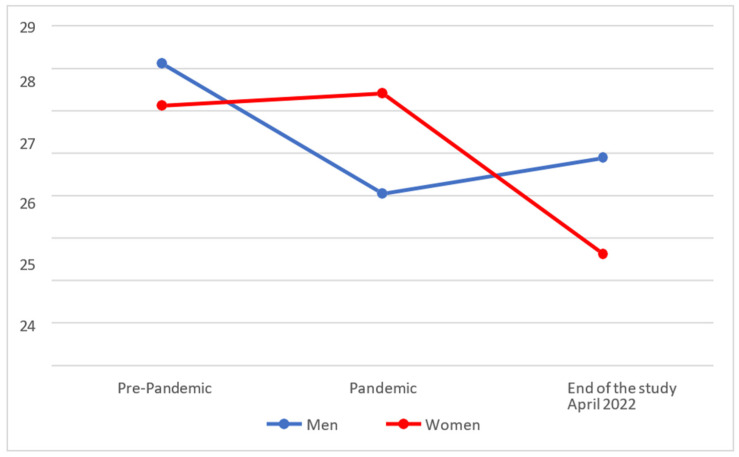
Evolution of the regulation factor of Emotional Intelligence.

**Figure 5 behavsci-14-00173-f005:**
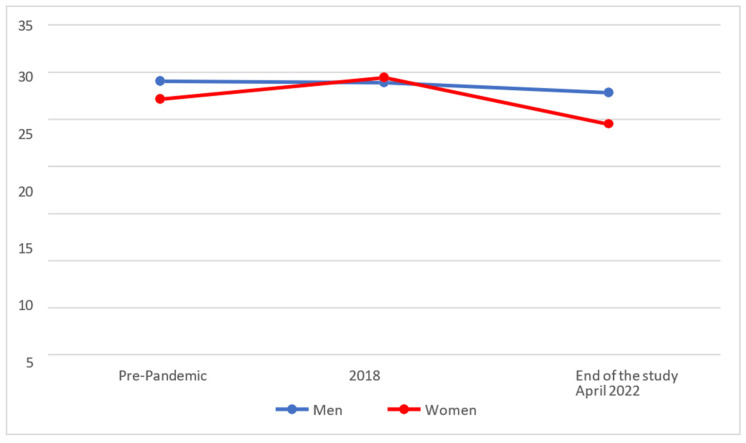
Evolution of the Self-Esteem factor of Effective Personality.

**Figure 6 behavsci-14-00173-f006:**
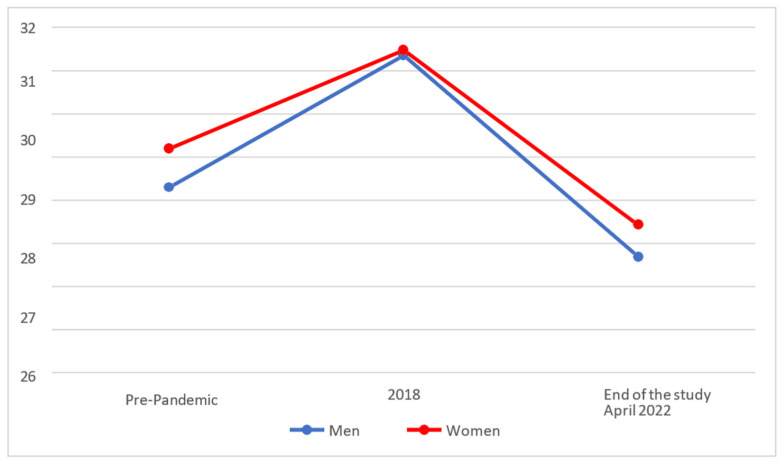
Evolution of the Academic Self-Realisation factor of Effective Personality.

**Figure 7 behavsci-14-00173-f007:**
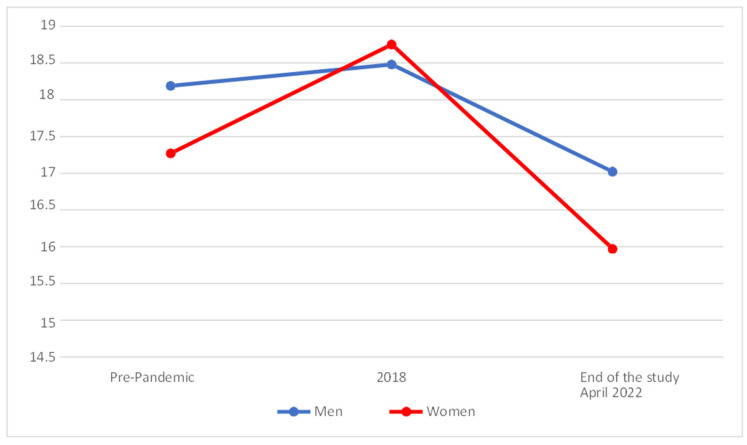
Evolution of the Resolute Self-Efficacy factor of Effective Personality.

**Figure 8 behavsci-14-00173-f008:**
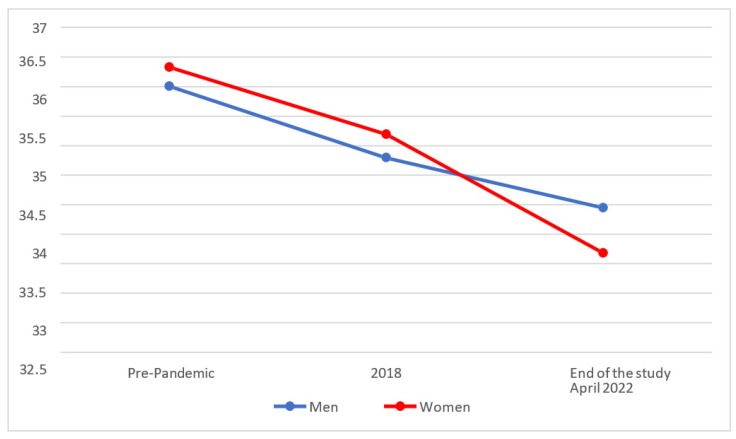
Evolution of the Social Self-Realisation factor of Effective Personality.

**Table 1 behavsci-14-00173-t001:** Summary of participants.

	Phase	Men	Women	Total
Anxiety	Pre-pandemic [47]	295	365	660
	Pandemic [50]	101	359	460
	Post-pandemic	49	262	311
Emotional Intelligence	Pre-pandemic [48]	75	280	355
	Pandemic [50]	32	59	91
	Post-pandemic	49	262	311
Effective Personality	Pre-pandemic [49]	210	498	708
	Pandemic [52]	25	149	174
	Post-pandemic	49	262	311

**Table 2 behavsci-14-00173-t002:** Spearman Rho correlations between the variables.

		State Anxiety	EI Attention	EI Clarity	EI Regulation	Self-Esteem	Academic Self- Realisation	Resolute Self-Efficacy	Social Self-Realisation
State Anxiety	Men	1.000	−0.088	−0.557 **	−0.422 **	−0.484 **	−0.304 *	−0.506 **	−0.579 **
	Women	1.000	0.100	−0.350 **	−0.319 **	−0.538 **	−0.402 **	−0.322 **	−0.456 **
EI attention	Men		1.000	0.538 **	-0.068	0.254	0.171	0.219	0.144
	Women		1.000	0.316 **	0.114	0.095	0.250 **	0.176 **	0.006
EI clarity	Men			1.000	0.260	0.540 **	0.194	0.556 **	0.449 **
	Women			1.000	0.383 **	0.551 **	0.480 **	0.468 **	0.270 **
EI regulation	Men				1.000	0.457 **	0.121	0.314 *	0.358 *
	Women				1.000	0.542 **	0.362 **	0.595 **	0.319 **
Self-Esteem	Men					1.000	0.385 **	0.502 **	0.488 **
	Women					1.000	0.584 **	0.555 **	0.477 **
Academic Self-Realisation	Men						1.000	0.544 **	0.094
	Women						1.000	0.509 **	0.348 **
Resolute Self-Efficacy	Men							1.000	0.341 *
	Women							1.000	0.258 **
Social Self-Realisation	Men								1.000
	Women								1.000

EI = Emotional Intelligence. * = Correlation is significant at the 0.05 level (bilateral). ** = Correlation is significant at the 0.01 level (bilateral).

## Data Availability

All data sets used in this study can be accessed through a request to the authors (cdi@ubu.es).

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
