# Peer review of "Evolution of Anxiety, Emotional Intelligence, and Effective Personality after COVID-19 among Spanish University Students"

_behavsci, 2024, doi:10.3390/bs14030173_

Round 1
Reviewer 1 Report
Comments and Suggestions for Authors
Dear Authors,
Very good paper. Please take the following into consideration:
1. The introduction is too long. Please divide it into introduction and Literature review. This way, your objectives come at the very beginning so that the reader knows what to expect.
2. Separate the conclusions for the discussions. The conclusions are too short. And also put limitations.
Comments on the Quality of English LanguagePlease write COVID-19.
Author Response
AUTHORS RESPONSES TO REVISOR 1
Dear Authors, Very good paper. Please take the following into consideration:
- The introduction is too long. Please divide it into introduction and Literature review. This way, your objectives come at the very beginning so that the reader knows what to expect.
AUTHORS RESPONSE: Our thanks to the Reviewer for such an interesting suggestion to improve the introduction. As indicated, we have divided it into an introduction and a literature review. We hope that having our objectives at the beginning will help the reader know what to expect.
- Separate the conclusions for the discussions. The conclusions are too short. And also put limitations.
AUTHORS RESPONSE: Thank you very much for your suggestion for improvement. As indicated, we have divided section 4. Conclusions and Discussion into two sections. We have expanded the conclusions. Regarding the limitations, they are indicated in the manuscript.
“As regards the objectives set out in this study, we can firstly conclude that although anxiety values have significantly increased during COVID-19, they have returned to pre-pandemic values among men, but have remained higher among women.
In second place, an increase in women's scores under all dimensions of Emotional Intelligence was observed during the pandemic. However, that increase has since reverted to even lower values at the time of this study than the pre-pandemic ones, except for emotional attention that has continued to increase. Regarding men, a slight, yet insignificant increase in emotional attention scores has been observed over time. Even so, they decreased in the other two dimensions during the pandemic. That decrease was maintained for emotional clarity, but has since been reversed, reaching pre-pandemic values for emotional regulation.
Thirdly, the scores for Effective Personality had a disparate evolution. Regarding Self-esteem, it was observed that the self-esteem of men was maintained at similar levels throughout all the periods, while women had significantly decreased their scores after the pandemic In Academic Self-Realization and Resolute Self-Efficacy, an improvement in scores was observed in 2018 and a considerable decrease after the pandemic, among both men and women. In relation to Social Self-Realization, a progressive decrease occurred in both sexes.
And finally, it was found that anxiety will be negatively correlated with the factors of Emotional Intelligence and Effective Personality; while Emotional Intelligence will be positively correlated with Effective Personality. With the exception of the dimension of emotional attention, which does not correlate with anxiety nor, in the group of men, with Effective Personality.
As regards the objectives set out in this study, we can conclude firstly that although anxiety values have increased significantly during COVID-19, they have returned to pre-pandemic values among men but have remained higher among women.
In second place, an increase in women's scores in all dimensions of Emotional Intelligence was observed during the pandemic. But this increase has reverted to values even lower than pre-pandemic at the time of this study, except for emotional attention, which has continued to increase. Regarding men, a slight, but not significant, increase in emotional attention scores has been seen over time. Despite this, they have experienced a decrease during the pandemic in the other two dimensions. This decrease has been maintained in emotional clarity but has been reversed, reaching values similar to pre-pandemic levels for in emotional regulation.
Thirdly, the levels of Effective Personality had a disparate evolution. Regarding Self-esteem, it was observed that men have maintained it in a similar way during all periods, while women, after the pandemic, have significantly decreased their scores. In Academic Self-Realization and Resolute Self-Efficacy, an improvement in scores was observed in 2018 and a considerable decrease after the pandemic, among both men and women. In relation to the Social Self-Realization, a progressive decrease occurred in both sexes.
And finally, it was found that anxiety was negatively correlated with the factors of Emotional Intelligence and Effective Personality; while Emotional Intelligence was positively correlated with Effective Personality. The dimension of emotional attention was an exception, which was neither correlated with anxiety nor, among the group of men, with Effective Personality.”
- Comments on the Quality of English Language. Please write COVID-19.
AUTHORS RESPONSE: We are grateful for that suggestion and have written COVID-19
Reviewer 2 Report
Comments and Suggestions for Authors
Dear Editor,
Thank you for the opportunity to review the manuscript entitled “Evolution of Anxiety, Emotional Intelligence and Effective Personality after COVID-19 among Spanish University Students”.
The aime of manuscript is interesting, however, some critical points emerge. In this regard, I have included some suggestions which I list below.
INTRODUCTION:
- Line 40: instead of "has stated" it is better to write "has reiterated", because the WHO had already given this definition of health years before...
- Line 43: instead of "the following were selected for the present work:"A better form might be "it seems particularly useful to analyze the following:"
- Line 44 and 45: remove "(Emotional attention, Clarity of feelings, Emotional)" and"(Self-Esteem, Academic Self-Realization, Resolute Self-efficacy, Social Self-Realization)" from here for the moment and subsequently describe (starting from line 47) the characteristics of the constructs, also trying to better link them to each other.
- It is necessary to better define the construct of emotional intelligence by making the distinction between this and that of Trait Emotional Intelligence, because the authors have administered an instrument that makes specific reference to the latter construct which, in recent years, has been particularly studied in the literature (see for example: Filippello, P., Sorrenti, L., Buzzai, C., & Costa, S. (2018). Predicting risk of school refusal: Examining the incremental role of trait EI beyond personality and emotion regulation. PSIHOLOGIJA, 2018, Vol. 51(1), 51–67; Gilles, P. Y., & Bailleux, C. (2001). Personality traits and abilities as predictors of academic achievement. European Journal of Psychology of Education, 16, 3–15; Mavroveli, S., & Sánchez-Ruiz, M. J. (2011). Trait emotional intelligence influences on academic achievement and school behaviour. British Journal of Educational Psychology, 81, 112–134). Therefore it appears necessary to detail the construct of Trait Emotional Intelligence, in order to create coherence between the introduction and methodology of the study carried out.
- The constructs described from line 47 to line 105 must be better connected to each other... the description appears disconnected and the reading is not harmonious.
METHODS AND MATERIALS
- The procedure for recruiting participants in the different phases is not clear...were they different participants? Did the authors draw on different studies in the literature? the instruments were therefore not administered to the same participants in the different phases (longitudinal study), or am I wrong? Perhaps it would be better to specify the procedure well.
Author Response
AUTHORS RESPONSES TO REVISOR 2
Dear Editor,Thank you for the opportunity to review the manuscript entitled “Evolution of Anxiety, Emotional Intelligence and Effective Personality after COVID-19 among Spanish University Students”.
The aime of manuscript is interesting, however, some critical points emerge. In this regard, I have included some suggestions which I list below.
INTRODUCTION:
- Line 40: instead of "has stated" it is better to write "has reiterated", because the WHO had already given this definition of health years before... -
AUTHORS RESPONSE: We are grateful to the Reviewer for that contribution and have included it in the manuscript.
- Line 43: instead of "the following were selected for the present work:"A better form might be "it seems particularly useful to analyze the following:"
AUTHORS RESPONSE: Our thanks to the Reviewer for that contribution, we have included it in the manuscript.
- Line 44 and 45: remove "(Emotional attention, Clarity of feelings, Emotional)" and"(Self-Esteem, Academic Self-Realization, Resolute Self-efficacy, Social Self-Realization)" from here for the moment and subsequently describe (starting from line 47) the characteristics of the constructs, also trying to better link them to each other.-
AUTHORS RESPONSE: Our thanks to the Reviewer for that suggestion. We have changed the order of the paragraph in which the concepts appear that you have suggested we remove. We have joined this paragraph with the following paragraphs discussing the characteristics of the constructs. We think this makes it easier to understand.
- It is necessary to better define the construct of emotional intelligence by making the distinction between this and that of Trait Emotional Intelligence, because the authors have administered an instrument that makes specific reference to the latter construct which, in recent years, has been particularly studied in the literature (see for example: Filippello, P., Sorrenti, L., Buzzai, C., & Costa, S. (2018). Predicting risk of school refusal: Examining the incremental role of trait EI beyond personality and emotion regulation. PSIHOLOGIJA, 2018, Vol. 51(1), 51–67; Gilles, P. Y., & Bailleux, C. (2001). Personality traits and abilities as predictors of academic achievement. European Journal of Psychology of Education, 16, 3–15; Mavroveli, S., & Sánchez-Ruiz, M. J. (2011). Trait emotional intelligence influences on academic achievement and school behaviour. British Journal of Educational Psychology, 81, 112–134). Therefore it appears necessary to detail the construct of Trait Emotional Intelligence, in order to create coherence between the introduction and methodology of the study carried out.
AUTHORS RESPONSE: We are grateful to the Reviewer for that contribution and have better defined the construct of emotional intelligence.
“[The Emotional Intelligence is defined by Salovey & Mayer (1989) as “the ability to monitor one's own and others' feelings and emotions, to discriminate among them, and to use this information to guide one's thinking and actions" (p. 187). Extremera (2020) notes that most emotionally intelligent individuals develop more effective strategies for dealing with those disorders, and react less intensely to stress, seeking adequate means of coping with it. In this context, Aguirre (2020) pointed out that the training of emotional skills “is a tool that enables one to attain new learning experiences that assist in the solution of specific situations with the application of knowledges, attitudes, values, skills, abilities and modes of behaviour” (p. 2). This study is based on the construct of Emotional Trait Intelligence. Emotional Trait Intelligence encompasses behavioral tendencies and self-perceptions related to the ability to identify, process and use emotionally charged information within various contexts remaining aware of individual differences (Petrides, et al., 2007; Peña-Sarrionandia, et al., 2015). Authors such as Joseph & Newman (2010), Filippello, et al. (2018) supported the idea that Emotional Trait Intelligence is conceptually different from Emotional Intelligence. The latter is considered as a series of cognitive skills related to emotions that imply capacity, while a trait refers to the self-perceptions of a person’s own emotional abilities, which is why they involve different processes (O'Boyle, et al., 2011; Qualter, et al., 2011; Van Rooy, et al., 2005).]”
“[Petrides, K. V., Pita, R., & Kokkinaki, F. (2007). The location of trait emotional intelligence in personality factor space. British Journal of Psychology, 98, 273–289. doi:10.1348/000712606X120618
Peña-Sarrionandia, A., Mikolajczak, M., & Gross, J. J. (2015). Integrating emotion regulation and emotional intelligence traditions: a meta-analysis. Frontiers in psychology, 6, 1–27. doi:10.3389/fpsyg.2015.00160
Joseph, D. L., & Newman, D. A. (2010). Emotional intelligence: an integrative meta-analysis and cascading model. Journal of Applied Psychology 95(1), 54–78. doi:10.1037/a0017286
Filippello, P., Sorrenti, L., Buzzai, C., & Costa, S. (2018). Predicting risk of school refusal: Examining the incremental role of trait EI beyond personality and emotion regulation. PSIHOLOGIJA, 51(1), 51–67
O’Boyle, E. H., Humphrey, R. H., Pollack, J. M., Hawver, T. H., & Story, P. A. (2011). The relation between emotional intelligence and job performance: A meta‐analysis. Journal of Organizational Behavior, 32(5), 788–818.
Qualter, P., Barlow, A., & Stylianou, M. S. (2011). Investigating the relationship between trait and ability emotional intelligence and theory of mind. British Journal of Developmental Psychology, 29(3), 437–454.
Van Rooy, D. L., Viswesvaran, C., & Pluta, P. (2005). An evaluation of construct validity: what is this thing called emotional intelligence?. Human Performance, 18(4), 445–462.]”
- The constructs described from line 47 to line 105 must be better connected to each other... the description appears disconnected and the reading is not harmonious
AUTHORS RESPONSE: We are grateful to the Reviewer for the suggested improvement, and the different constructs have been more coherently linked together.
METHODS AND MATERIALS
- The procedure for recruiting participants in the different phases is not clear...were they different participants? Did the authors draw on different studies in the literature? the instruments were therefore not administered to the same participants in the different phases (longitudinal study), or am I wrong? Perhaps it would be better to specify the procedure well..
AUTHORS RESPONSE: We are grateful to the Reviewer for such an interesting suggestion to improve the Procedure section. Following the above indications, we have completed this section. We hope that it is now suitable.
“This study was developed during the second semester of the 2021-2022 academic course. After reviewing the literature, the aims, the hypotheses, and instruments of the study were defined. Subsequently, the articles describing pre-pandemic and pandemic phases were selected, based on the previously mentioned participant selection criteria. Finally, the questionnaire was administered to the set of Spanish universities to obtain the post-pandemic phase participants. The questionnaires were applied on a voluntary basis in the Microsoft Forms format….”
Reviewer 3 Report
Comments and Suggestions for Authors
The evaluated manuscript focuses on the evolution of anxiety, emotional intelligence, and effective personality in university students from Spain after the COVID-19 pandemic. The article's topic is relevant and current, and the results related to the three variables under evaluation allow for drawing conclusions of interest for the fields of educational and psychological intervention. Despite these strengths, it is important to note certain deficiencies in the manuscript that need correction:
Introduction:
The introduction is coherent, complete, and well-structured. However, a more detailed explanation about the comparison between genders regarding the three evaluated variables is recommended. Currently, only a brief mention is made. Given that the subsequent results reveal the evolution by gender, the comparison between men and women should be included as an objective of the study, posing a hypothesis in this regard.
Furthermore, while a review of works analyzing anxiety, emotional intelligence, and personality in the general population is provided, it is necessary to adapt this explanation more extensively to the university population.
Method:
Has authorization for the study been requested from any Ethics Committee? This should be reported.
It is recommended to create an image or table that shows participants in the three phases in an integrated manner. Additionally, specify the courses to which participants belong within the selected titles.
Present the score ranges for each of the TMMS-24 dimensions, as well as the reliability coefficients for each scale.
Results:
Include results based on an objective related to gender differences.
Discussion:
Similar to the introduction, the discussion should delve into gender differences concerning the three study variables.
Formal Aspects:
There are some typos in the writing, as well as different font sizes that should be carefully reviewed. Adjust the references to the APA 7th edition format, addressing issues such as the inconsistent indication of dois in all references. Finally, it is recommended that the entire manuscript be reviewed for English, ideally by a native speaker.
Comments on the Quality of English LanguageThe writing is, in general, correct and fluid. However, it is recommended that the entire manuscript be reviewed for English, ideally by a native speaker.
Author Response
AUTHORS RESPONSES TO REVISOR 3
Comments and Suggestions for Authors
The evaluated manuscript focuses on the evolution of anxiety, emotional intelligence, and effective personality in university students from Spain after the COVID-19 pandemic. The article's topic is relevant and current, and the results related to the three variables under evaluation allow for drawing conclusions of interest for the fields of educational and psychological intervention. Despite these strengths, it is important to note certain deficiencies in the manuscript that need correction:
INTRODUCTION:
The introduction is coherent, complete, and well-structured. However, a more detailed explanation about the comparison between genders regarding the three evaluated variables is recommended. Currently, only a brief mention is made. Given that the subsequent results reveal the evolution by gender, the comparison between men and women should be included as an objective of the study, posing a hypothesis in this regard.
AUTHORS RESPONSE: Our thanks to the Reviewer for the interest expressed in the study and the suggestions for its improvement. On the one hand, a more detailed explanation has been added on the comparison between genders regarding the three evaluated variables. On the other hand, it should be noted that the hypotheses proposed for the three variables refer to the differences between men and women. Consequently, and following your suggestion, it has also been added to the objectives.
“In relation to gender and different levels of anxiety in the university population, several authors have indicated a different rate of maladaptive anxiety between 23-25% in men and between 36-42% in women (Agudelo-Vélez et al., 2008; Bayram & Bilgel, 2008; Doumit et al., 2017). A difference that is two to three times higher than the general population (8 % and 15 % respectively) (Andrews, Hejdenberg & Wilding, 2006). Although others found no such differences (Medina et al., 2019).”
Agudelo-Vélez, D. M., Casadiegos-Garzón, C. P., & Sánchez-Ortíz, D. L. (2008). Características de ansiedad y depresión en estudiantes universitarios. International Journal of Psychological Research, 1(1), 34-39
Bayram, N., & Bilgel, N. (2008). The prevalence and sociodemographic correlations of depression, anxiety and stress among a group of university students. Social Psychiatry and Psychiatric Epidemiology, 43(8), 667-672. doi: 10.1007/ s00127-008-0345-x
Doumit, R., Khazen, G., Katsounari, I., Kazandjian, C., Long, J., & Zeeni, N. (2017). Investigating vulnerability for developing eating disorders in a multiconfessional population. Community Mental Health Journal, 53(1), 107-116. doi: 10.1007/s10597-015-9872-6
Medina-Gómez, M.B., Martínez Martín, M.A., Escolar-Llamazares, M.C., González-Alonso, Y. & Mercado-Vale, E. (2019). Ansiedad e insatisfacción corporal en universitarios. Acta Colombiana de Psicología 22 (1): 13-21
“In relation to gender-based differences and Effective Personality, authors such as Castellanos et al. (2014) indicated that women have higher scores in Academic Self-Actualization; while men show better Self-Esteem and Resolutional Self-Efficacy. While other studies such as that of Vargas (2017) found no such differences among the university population”.
Castellanos, S., Guerra, P. y Bueno, J.A. (2014). Diferencias de género en personalidad eficaz en universitarios chilenos. International Journal of Developmental and Educational Pschology, 1(5), 131-140. https://doi.org/10.17060/ijodaep.2014.n1.v5.655
Vargas, C. (2017). Tipología modales multivariadas de personalidad eficaz y competencias emocionales en universitarios de República Dominicana [Doctoral Thesis]. Complutense University of Madrid.
Furthermore, while a review of works analyzing anxiety, emotional intelligence, and personality in the general population is provided, it is necessary to adapt this explanation more extensively to the university population.
AUTHORS RESPONSE: Our thanks for the suggested improvement. We have adapted the explanation of the three constructs to the university population.
“Emotional Intelligence has been extensively studied in university populations. In the study of Fragoso (2018), almost half of the students needed to improve their emotional attention skills (45%), 42% emotional clarity and 31% emotional repair. The Emotional Intelligence of university students is also related to different variables such as the studies that relate it to coping strategies for stress (Puigbó et al., 2019), negative affect related to suicide among young people (Gómez-Romero et al., 2018), and performance (Arntz and Trunce, 2019).”
Fragoso, L. R. (2018). Inteligencia emocional en estudiantes de educación superior. análisis a través de técnicas mixtas. International Journal of Developmental and Educational Psychology, (2) 231-240. https://www.redalyc.org/articulo.oa?id=349857778020
Puigbó, J., Edo, S., Rovira, T., Limonero, J. T. y Fernández-Castro, J. (2019). Influencia de la inteligencia emocional percibida en el afrontamiento del estrés cotidiano. Revista Ansiedad y Estrés, 25(1). 1-16. https://doi.org/10.1016/j.anyes.2019.01.003
Gómez-Romero, M. J., Limonero, J. T., Trallero, T. J., Montes-Hidalgo, J. y TomásSábado, J. (2018). Relación entre inteligencia emocional, afecto negativo y riesgo suicida en jóvenes universitarios. Revista Ansiedad y Estrés, 24(1) 18-23. https://doi.org/10.1016/j.anyes.2017.10.007
Arntz, V. J. y Trunce, M. S. (2019). Inteligencia emocional y rendimiento académico en estudiantes universitarios de nutrición. Investigación en Educación Médica, 8(31),82-91. DOI: 10.22201/facmed.20075057e.2019.31.18130
METHOD:
Has authorization for the study been requested from any Ethics Committee? This should be reported.
AUTHORS RESPONSE: The study was performed in accordance with the ethical standards as laid down in the 1964 Declaration of Helsinki and its later amendments or comparable ethical standards. Furthermore, The students were informed of the objectives and the procedures of the study, as well as the anonymized treatment of the data. They were also informed that their participation would be voluntary and that they could abandon the research any time without any prejudices. All participating students signed an informed consent. All data collected for this study have undergone systematic anonymization to prevent any personal identification. The anonymization protocol used has been carried out following the instructions of the Ethics Committee of the University of Burgos. We have added this information in the manuscript
It is recommended to create an image or table that shows participants in the three phases in an integrated manner. Additionally, specify the courses to which participants belong within the selected titles.
AUTHORS RESPONSE: Thank you very much for the suggestion for improvement. We have proceeded to incorporate the table as suggested.
“In summary, Table 1 presents the data on the participants for both men and women in each phase (Table 1).”
Table 1
Summary of participants
Present the score ranges for each of the TMMS-24 dimensions, as well as the reliability coefficients for each scale.
AUTHORS RESPONSE: Our thanks for the suggestion. We have proceeded to incorporate the information as suggested.
“Emotional Regulation/Repair. This instrument presents a Cronbach’s alpha reliability of 0.90. The internal consistency for each subscale was over.85 in all cases, and the test-retest correlations after 4 weeks ranged from .60 to .83. The items are scored on a five-point Likert-type scale, ranging from 1=zero agreement to 5=complete agreement, so the score ranges of each dimension were between one and forty.”
RESULTS:
Include results based on an objective related to gender differences.
AUTHORS RESPONSE: Ours thanks to the reviewer you the suggestion. We have expanded the conclusions section where the information that was requested for each of the gender-based objectives can be seen.
DISCUSSION:
Similar to the introduction, the discussion should delve into gender differences concerning the three study variables.
AUTHORS RESPONSE: Thank you again. Similarly, in the Introduction section, we have included gender-based literature for each of the three constructs.
FORMAL ASPECTS:
There are some typos in the writing, as well as different font sizes that should be carefully reviewed. Adjust the references to the APA 7th edition format, addressing issues such as the inconsistent indication of dois in all references. Finally, it is recommended that the entire manuscript be reviewed for English, ideally by a native speaker.
AUTHORS RESPONSE: Ours thanks once again to the Reviewer for the interest in improving our manuscript. Following the above advice, we have reviewed and corrected all typos in the writing, the different font sizes, adjust the references to the APA 7th edition format, and a native English speaker has reviewed the entire manuscript.
Comments on the Quality of English Language. The writing is, in general, correct and fluid. However, it is recommended that the entire manuscript be reviewed for English, ideally by a native speaker..
AUTHORS RESPONSE: Again, our thanks to the Reviewer for the suggested improvement. A native English speaker has reviewed the entire manuscript.